ⓐ | **Open Peer Review** | Mycoplasmology | Research Article

# Peripheral blood inflammatory cytokines linked to clinical outcomes in *Mycoplasma pneumoniae* pneumonia

Yuling Bao,[1] Chang Shu,[1] Yan Liu,[1] Anni Cao,[1] Rui Zhang,[1] Deyu Zhao,[1] Feng Liu[1]

**ABSTRACT** *Mycoplasma pneumoniae* pneumonia (MPP) can result in a variety of poor prognoses. A retrospective cohort study of MPP patients admitted to the Children's Hospital of Nanjing Medical University from 1 January 2021 to 1 April 2024 was carried out to evaluate the association between 12 peripheral blood inflammatory cytokines and clinical outcomes. 2,391 patients were enrolled and divided into four outcome-based groups: severe MPP (SMPP), refractory MPP (RMPP), necrotizing pneumonia (NP), and pulmonary embolism (PE). The levels of interleukin-2 (IL-2), interleukin-4 (IL-4), interleukin-10 (IL-10), and interleukin-17 were elevated in the poor-prognosis groups of SMPP and RMPP. Interferon-γ (IFN-γ) was upregulated in the poor-prognosis groups of RMPP, and it was an influencing factor for the prognoses. Interleukin-6 (IL-6) was upregulated in the NP and PE groups, and it was an influencing factor for NP. Correlation analysis showed strong positive correlations among the 12 inflammatory variables. IFN-γ, IL-6, and IL-10 were positively correlated with C-reactive protein, neutrophil-to-lymphocyte ratio, D-dimer, and lactate dehydrogenase. These findings suggest that inflammatory cytokines may serve as potential indicators for predicting different prognoses of MPP.

**IMPORTANCE** The number of patients in our clinical research cohort is the largest to date in studies focusing on Mycoplasma pneumonia (MPP) and inflammatory cytokines. Specifically, we established a prospective cohort study involving 2,391 patients, systematically collecting clinical data and peripheral blood samples to explore the association between peripheral blood inflammatory cytokine levels and clinical outcomes of MPP. Our findings hold substantial clinical significance: patients with poor MPP outcomes exhibited a markedly stronger pulmonary immunoinflammatory response, shedding new light on the underlying mechanisms driving disease progression. Importantly, incorporating cytokine profiles into routine clinical monitoring and early diagnosis could facilitate timely identification and targeted management of adverse MPP prognoses.

**KEYWORDS** inflammatory cytokine, *Mycoplasma pneumoniae* pneumonia, clinical outcomes

M ycoplasma pneumoniae pneumonia (MPP), an inflammatory pulmonary condition triggered by *Mycoplasma pneumoniae* (MP) infection, significantly contributes to lower respiratory tract infections in children. Predominantly affecting school-age children and adolescents, MPP usually reaches its peak incidence during spring and autumn. It often occurs in clusters and accounts for approximately 30–40% of community-acquired pneumonia (CAP) cases (1). In recent years, there has been an increase in the incidence of severe MPP (SMPP) and refractory MPP (RMPP), giving rise to complications such as pleural effusion, pulmonary atelectasis, plastic bronchitis, necrotizing pneumonia (NP), and pulmonary embolism (PE). Moreover, MPP can cause a wide range of extrapulmonary manifestations, including hepatitis, myocarditis, rash, encephalitis, hemolytic

**Peer Reviewers** Ahmed Adel Baz, Lanzhou Veterinary Research Institute, Assiut, Zimbabwe; Robert Doug Hardy, ID Specialists, Plano, Texas, USA

Address correspondence to Feng Liu, axsliu@njmu.edu.cn, or Deyu Zhao, zhaodeyu98@126.com.

The authors declare no conflict of interest.

See the funding table on p. 9.

anemia, arthritis, and Guillain–Barré syndrome. These manifestations may have long-term impacts on small airway ventilation function, thereby imposing a substantial medical burden on the treatment of pediatric MPP (2–4).

The pathogenesis of MPP remains elusive. Pro-inflammatory cytokines, including tumor necrosis factor (TNF-α), interleukin-1β (IL-1β), interleukin-4 (IL-4), interleukin-6 (IL-6), and interleukin-8 (IL-8), exhibit upregulation (5, 6). In the bronchoalveolar lavage fluid (BALF) of MPP patients, the levels of IL-6, IL-8, interleukin-10 (IL-10), interferon-γ (IFN-γ), and TNF-α are elevated. Changes in cytokine levels in serum have also been detected, with the trend of inflammatory cytokine changes consistent with that in BALF (7). The levels of IL-4 and the ratio of IL-4/IFN-γ in the BALF of MPP patients are significantly increased, indicating that the T-helper 2 (Th2) immune response is predominant (8). Our previous studies revealed that the MP load on the focal side of MPP is not higher than that on the healthy side, and the severity of MPP is associated with neutrophil infiltration, which supports the crucial role of immune responses in the pathogenesis of MPP (9). Therefore, it is imperative for us to investigate the relationship between peripheral blood inflammatory cytokines and the clinical outcomes of MPP.

To address this, we conducted a 4-year study involving 2,391 patients to investigate the relationship between inflammatory cytokines and different prognoses of MPP. Our aim was to provide a basis for predicting MPP prognosis and to elucidate its pathogenesis by examining the correlation between peripheral blood inflammatory cytokine levels and MPP prognoses.

## RESULT

### Patient characteristics and laboratory findings

The clinical characteristics of patients in the SMPP and RMPP outcome-based groups are shown in Table 1. The clinical characteristics of children with the NP and PE groups are provided in Table S1A and B online. The SMPP and RMPP groups had a higher proportion of females (51.7% and 52.6%, $P < 0.01$), an older age at onset (7.17 years and 7 years, $P < 0.01$), and a longer pre-admission fever duration (7 days and 8 days, $P < 0.01$). Additionally, the SMPP and RMPP groups exhibited higher levels of C-reactive protein (CRP), white blood cells (WBC), platelets, neutrophil-to-lymphocyte ratio (NLR), lactate dehydrogenase (LDH), and D-dimer, and a higher incidence of pleural effusion ($P < 0.05$). The pre-hospital fever duration, levels of WBC, LDH, NLR, and D-dimer, and incidence of pleural effusion were all significantly higher in the NP group than in the non-NP group. Similarly, in the PE group, the pre-hospital fever duration, CRP, LDH, NLR, and D-dimer levels, along with the incidence of pleural effusion, were all significantly higher than in the non-PE group. Among the four groups, the poor-prognosis groups had lower activated partial thromboplastin time (APTT) levels.

### Analysis of inflammatory cytokines

Inflammatory cytokine data are presented in Table S2A through D (available online). Logistic regression analyses, with SMPP, RMPP, PE, and NP as the dependent variable in each analysis, were performed to identify factors associated with disease progression after adjustment for age and sex. The results are presented in Table 2 and Fig. 1A through E. The SMPP group had lower levels of IL-1β (IQR: 11.58, $P < 0.01$), IL-5 (IQR: 2.27, $P < 0.01$), and IL-8 (IQR: 5.35, $P < 0.01$) and higher levels of IL-2 (IQR: 1.78, $P < 0.01$), IL-4 (IQR: 1.16, $P < 0.05$), IL-10 (IQR: 2.0, $P < 0.05$), and IL-17 (IQR: 4.88, $P < 0.05$); logistic regression analysis indicated that IL-4 (OR = 1.334, 95% CI: 1.176–1.513, $P < 0.01$), IL-5 (OR = 0.987, 95% CI: 0.976–0.998, $P < 0.05$), and IL-1β (OR = 0.986, 95% CI: 0.981–0.992, $P < 0.001$) are influencing factors for SMPP. The RMPP exhibited significantly higher levels of IFN-γ (IQR: 13.12, $P < 0.01$), IL-2 (IQR: 1.78, $P < 0.01$), IL-4 (IQR: 1.16, $P < 0.01$), IL-10 (IQR: 2.05, $P < 0.01$), and IL-17 (IQR: 5.05, $P < 0.01$); IFN-γ (OR = 1.003, 95% CI: 1.001–1.005, $P < 0.01$) is an influencing factor for RMPP. The PE group had lower levels of IL-1β (IQR: 9.49, $P < 0.05$), IL-5 (IQR: 1.72, $P < 0.05$), IL-8 (IQR: 1.56, $P < 0.01$), and IL-12p70 (IQR: 1.25, $P < 0.05$).

**TABLE 1** Admission characteristics of children with SMPP and RMPP in the retrospective cohort according to their subsequent clinical outcome

| A SMPP | | | | |
|---|---|---|---|---|
| Characteristic | total MPP (n = 2,391) | SMPP (n = 1,824) | Non-SMPP (n = 567) | P-value |
| Age (years) | 7.03 (4.94, 8.67) | 7.17 (5.31, 8.77) | 6.49 (3.96, 8.33) | <0.001 |
| Sex, n (%) | | | | |
| Male | 1,186 (49.60) | 881 (48.30) | 305 (53.79) | <0.001 |
| Female | 1,205 (50.40) | 943 (51.70) | 262 (46.21) | |
| Fever, n (%) | 2,275 (95.15) | 1,759 (96.44) | 516 (91.01) | <0.001 |
| Preadmission fever duration (d) | 7.00 (5.00, 10.00) | 7.00 (5.00, 10.00) | 5.00 (3.00, 7.00) | <0.001 |
| CRP, mg/L | 7.19 (2.61, 20.45) | 8.02 (2.95, 22.71) | 5.21 (2.15, 13.05) | <0.001 |
| WBC, $\times 10^9$/L | 9.66 (7.17, 12.86) | 9.99 (7.39, 13.21) | 8.67 (6.64, 11.64) | <0.001 |
| Hemoglobin, g/L | 126.00 (119.00, 132.00) | 126.00 (119.00, 132.00) | 126.00 (119.00, 132.00) | 0.916 |
| Platelets, $\times 10^9$/L | 319.00 (236.00, 419.00) | 323.00 (237.00, 426.00) | 307.00 (233.00, 394.00) | 0.041 |
| NLR, % | 246.69 (157.25,423.89) | 269.33 (165.77,447.65) | 208.79 (134.76,44.37) | <0.001 |
| LDH, U/L | 318.00 (270.00, 388.00) | 323.00 (273.75, 401.00) | 298.00 (262.00, 358.00) | <0.001 |
| PT, s | 12.00 (11.40, 12.90) | 12.00 (11.30, 12.90) | 12.10 (11.50, 12.90) | 0.343 |
| APTT, s | 31.10 (28.20, 34.30) | 30.80 (27.90, 33.80) | 32.40 (29.30, 35.50) | <0.001 |
| D-dimer, ng/mL | 226.00 (149.00, 400.00) | 242.00 (155.00, 455.00) | 192.00 (127.75, 290.25) | <0.001 |
| Fibrinogen, g/L | 3.34 (2.84, 3.80) | 3.34 (2.84, 3.80) | 3.35 (2.90, 3.78) | 0.881 |
| Pleural effusion, n (%) | 183 (7.65) | 166 (9.10) | 17 (3.00) | <0.001 |
| B RMPP | | | | |
| Characteristic | Total MPP (n = 2,391) | RMPP (n = 1,526) | Non-RMPP (n = 865) | P-value |
| Age (years) | 7.03 (4.94, 8.67) | 7.33 (5.65, 8.88) | 6.33 (3.93, 8.28) | <0.001 |
| Sex, n (%) | | | | |
| Male | 1,186 (49.60) | 724 (47.44) | 462 (53.41) | <0.001 |
| Female | 1,205 (50.40) | 802 (52.56) | 403 (46.59) | |
| Fever, n (%) | 2,275 (95.15) | 1,475 (96.66) | 800 (92.49) | <0.001 |
| Preadmission fever duration (d) | 7.00 (5.00, 10.00) | 8.00 (6.00, 10.00) | 6.00 (4.00, 7.00) | <0.001 |
| CRP, mg/L | 7.19 (2.61, 20.45) | 8.23 (2.85, 24.42) | 5.85 (2.34, 14.84) | <0.001 |
| WBC, $\times 10^9$ /L | 9.66 (7.17, 12.86) | 10.10 (7.47, 13.24) | 8.86 (6.86, 12.13) | <0.001 |
| Hemoglobin, g/L | 126.00 (119.00, 132.00) | 126.00 (119.00, 133.00) | 126.00 (119.00, 132.00) | 0.416 |
| Platelets, $\times 10^9$ /L | 319.00 (236.00, 419.00) | 325.00 (240.00, 427.00) | 306.00 (231.75, 402.00) | 0.018 |
| NLR, % | 246.69 (157.25, 423.89) | 272.89 (171.01, 454.17) | 211.01 (138.40, 359.08) | <0.001 |
| LDH, U/L | 318.00 (270.00, 388.00) | 324.00 (272.00, 410.00) | 308.00 (264.00, 363.00) | <0.001 |
| PT, s | 12.00 (11.40, 12.90) | 12.10 (11.40, 12.90) | 12.00 (11.30, 12.80) | 0.43 |
| APTT, s | 31.10 (28.20, 34.30) | 30.70 (27.80, 33.90) | 31.90 (29.20, 34.80) | <0.001 |
| D-dimer, ng/mL | 226.00 (149.00, 400.00) | 263.00 (164.00, 525.25) | 183.00 (132.00, 265.00) | <0.001 |
| Fibrinogen, g/L | 3.34 (2.84, 3.80) | 3.36 (2.83, 3.80) | 3.31 (2.86, 3.79) | 0.931 |
| Pleural effusion, n (%) | 183 (7.65) | 156 (10.22) | 27 (3.12) | <0.001 |

In contrast, both the PE and NP groups had higher levels of IL-6 (IQR: 9.07, $P < 0.01$ and IQR: 6.54, $P < 0.001$); the results indicated that IL-6 (OR = 1.008, 95% CI: 1.003–1.012, $P < 0.01$) is a significant factor influencing the progression of NP. Including age and sex in the analysis revealed similar results.

## Heat map clustering of 12 inflammatory variables across 4 outcomes Inflammatory cytokine distribution

For cytokine clustering, similarities among various cytokines (based on their expression profiles) were calculated using Euclidean distance, followed by hierarchical clustering with Ward's D2 algorithm to ultimately group similar cytokines together. Inflammatory responses in MPP are intricately associated with its prognosis. Through a cluster analysis of the association between inflammatory cytokines and prognosis, we have preliminarily discovered that the levels of most inflammatory cytokines in patients with poor prognosis are remarkably elevated. However, the specific inflammatory cytokines with increased levels vary according to different prognostic conditions. (Fig. 1F)

**TABLE 2** Cytokines significantly associated with four outcomes following logistic regression analysis[a]

|  | Variables | Crude OR (95% CI) | *P*-value | OR (95% CI)[b] | *P*-value |
|---|---|---|---|---|---|
| SMPP | IL-4 | 1.334 (1.176–1.513) | <0.001 | 1.321 (1.165–1.498) | <0.001 |
|  | IL-5 | 0.987 (0.976–0.998) | 0.017 | 0.987 (0.976–0.998) | 0.018 |
|  | IL-1β | 0.986 (0.981–0.992) | <0.001 | 0.987 (0.981–0.992) | <0.001 |
| RMPP | IFN-γ | 1.003 (1.001–1.005) | 0.018 | 1.003 (1.000–1.005) | 0.038 |
| NP | IL-6 | 1.008 (1.003–1.012) | <0.001 | 1.008 (1.004–1.012) | <0.001 |

[a]Inflammatory factors exhibiting significant within-group heterogeneity (*P* < 0.15) were included in the models. OR, odds ratio; CI, confidence interval; SMPP, severe MPP; RMPP, refractory MPP; NP, necrotizing pneumonia.
[b]Adjusted for potential confounders: age and sex.

## Analysis of the relationships among inflammatory cytokines CRP, LDH, D-dimer, and NLR

A further correlation analysis was conducted to examine both the relationships among the 12 inflammatory cytokines and those between each factor and CRP, LDH, D-dimer, and NLR. The results showed strong positive associations among each pair of the 12 inflammatory variables. CRP, NLR, D-dimer, and LDH were all positively correlated with IFN-γ, IL-6, and IL-10. IL-1β, IL-4, and interferon-α (IFN-α) were all negatively correlated with CRP. LDH was positively correlated with IL-5, IL-17, and IL-1β. D-dimer was positively correlated with IL-5, IL-10, and IFN-α (Fig. 2).

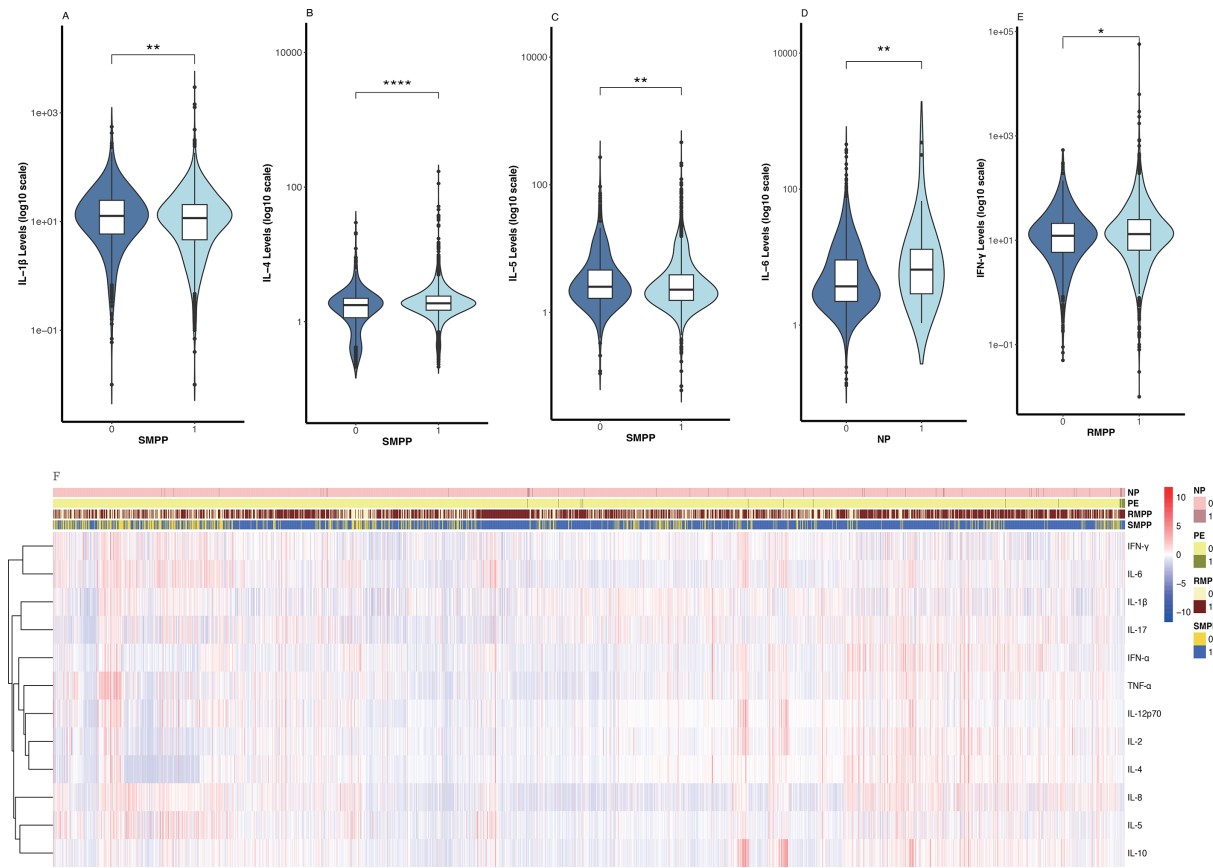

**FIG 1** Violin plots (A–E) show five inflammatory cytokines' concentration distributions in MPP patients stratified by distinct outcomes, with y-axes as log₁₀-scaled cytokine levels (reducing data skewness). (F) Clustered heatmap of 12 inflammatory cytokines across four clinical phenotypes: SMPP, RMPP, NP, and PE. Statistical analysis for clustering: Similarities among cytokine expression profiles were quantified by Euclidean distance and subjected to hierarchical clustering using Ward's D2 algorithm. *P < 0.05; **P < 0.01; ****P < 0.0001.

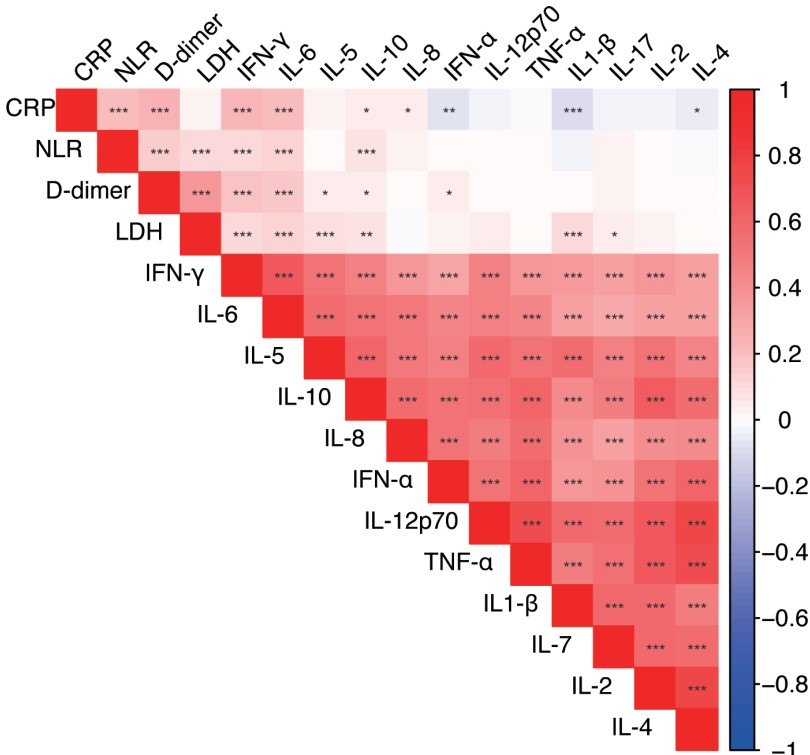

**FIG 2** Heatmap of the correlations between 12 inflammatory cytokines and CRP, NLR, D-dimer, and LDH. The bar represents the range of parameter correlations, ranging from −1 (dark blue) to 1 (dark red). The stronger the correlation, the larger the absolute value. A negative sign indicates a negative correlation. *$P$ < 0.05; **$P$ < 0.01; ***$P$ < 0.001.

## The association between prognosis and the quantity of abnormal inflammatory cytokines

The U-test was used to assess intergroup differences in the cumulative increase of inflammatory cytokines within each outcome group. The results revealed a statistically significant difference between the SMPP and non-SMPP groups ($P$ < 0.05) (Table 3). Nevertheless, no such difference was observed in the prognoses of RMPP, NP, and PE. The greater the number of abnormal inflammatory cytokines, the higher the likelihood of severe mycoplasma pneumonia.

### Value of cytokine assessment in evaluating the outcomes of MPP

The diagnostic value of cytokine assessment for the outcomes of MPP was evaluated using Receiver operating characteristic (ROC) analysis. The results revealed that the AUCs for IFN-γ, IL-4, IL-5, IL-1β, and IL-6 were 0.530, 0.578, 0.5, 0.536, and 0.607 (Fig. S1).

### Principal component analysis plot (PCA) and feature loadings

PCA identified two components explaining 75.3% (PC1: 52.4%, PC2: 22.9%) variance. PC1 had high loadings from IL-4, IL-1β, IL-17, TNF-α, and IFN-γ. PC2 had high loadings from IFN-α, IL-12p70, IL-5, IL-2, and IL-10. This indicates that the first two principal components are driven by different inflammatory factors, respectively (Fig. S2).

## DISCUSSION

To the best of our knowledge, this is the largest clinical study exploring the association between inflammatory cytokines and the outcomes of MPP. Our research shows that inflammatory cytokines are linked to several unfavorable prognoses of MPP, offering a

**TABLE 3** The link between the number of aberrant cytokines and four specific outcomes[a]

|  | Outcomes | M（P$_{25}$, P$_{75}$） | Z | *P*-value |
|---|---|---|---|---|
| SMPP | 0 | 2（1, 3） | −2.229 | 0.022 |
|  | 1 | 1（0, 3） |  |  |
| RMPP | 0 | 1（0, 3） | −1.175 | 0.24 |
|  | 1 | 2（1, 3） |  |  |
| NP | 0 | 2（0, 3） | −0.675 | 0.5 |
|  | 1 | 1（0.25, 2） |  |  |
| PE | 0 | 2（0, 3） | −0.94 | 0.347 |
|  | 1 | 2（1, 4） |  |  |

[a]Data are presented as median (interquartile range). SMPP, severe mycoplasma pneumoniae pneumonia; RMPP, refractory mycoplasma pneumoniae pneumonia; NP, necrotizing pneumonia; PE, pulmonary embolism.

foundation for predicting the disease's prognosis and elucidating its pathogenesis. A primary feature of MPP is widespread inflammation. The overproduction of pro-inflammatory cytokines is a factor that aggravates lung injury. This study investigated the relationship between peripheral blood inflammatory cytokines and outcomes of MPP.

The study revealed that children with poor prognoses in the RMPP group exhibited higher levels of IFN-γ. IFN-γ is a crucial pro-inflammatory cytokine involved in both innate and adaptive immunity. It can activate macrophages, combat infections, and notably contribute to lung injury (10, 11). The findings of our study align with those of several previous investigations, which have reported a significant increase in serum IFN-γ levels in RMPP cases (12). In our study, elevated serum IFN-γ levels were significant factors influencing the progression of RMPP.

Our study shows elevated IL-2 levels in the RMPP and SMPP groups. IL-2, secreted mainly by T-helper 1 (Th1) cells, is crucial for T-cell immune regulation. Some studies note rising IL-2 levels during mycoplasma infection and SMPP onset [12, 13, 14]. Others report lower IL-2 levels in SMPP than in general MPP (GMPP) [15]. However, our study found no significant reduction in IL-2. This discrepancy may be related to the patient's age, infection severity, or immune status. The inconsistent IL-2 changes across studies provide a basis for further exploring its role in MPP. IL-4 and IL-5 are largely produced by activated Th2 cells, contributing to humoral immunity and often linked to allergies and asthma. Studies indicate higher IL-4 levels in children with MPP than in those with Streptococcus pneumoniae infection or healthy controls (13). In MP-infected mice, IL-4 levels rose in BALF and splenocytes (11, 14). Our study found elevated IL-4 levels in the SMPP and RMPP groups. IL-4 was an influencing factor for poor SMPP prognosis, suggesting an immune imbalance in children. Current research on IL-5 focuses on its link with MP-induced wheezing or asthma. In wheezing children with MP infection, IL-5 concentrations were significantly higher in subjects with acute MPP infection than in those without such infection. Only IL-5 differed significantly from controls (15). Our study demonstrated that patients with SMPP and PE had lower serum IL-5 levels. This finding may be attributed to several factors: first, the older age of our cohort (predominantly school-aged children) and the more severe mycoplasma infections (SMPP, PE); second, IL-5 plays a significant role in eosinophilic inflammation, which may differ from its role in severe pulmonary parenchymal infectious diseases. Currently, no studies have elucidated the relationship between eosinophilic inflammation and mycoplasma infection. Additionally, the relatively low incidence of asthma in China and differences in genetic backgrounds may also contribute to this observation. Furthermore, although the level of IL-4 (a Th2-type cytokine) was elevated in the SMPP group, IL-5 levels were decreased, suggesting differential activation of the Th2 pathway (11).

In the poor-prognosis groups of PE and NP, the level of IL-6 was higher; IL-6, secreted by mononuclear macrophages, T and B lymphocytes, etc., initiates and amplifies the inflammatory response. Its high expression can cause significant immunological inflammatory harm as it promotes B-cell proliferation and differentiation, leading to the production of multiple IgE and IgG antibodies (16). Studies have shown that IL-6 levels

in the SMPP group are much higher than those in the MMPP group, suggesting that IL-6 overexpression is linked to the extent of pediatric lung damage (17). Our study showed that elevated IL-6 was an influencing factor for NP. This suggests that IL-6 may act as a marker of inflammatory response and be involved in the progression of NP.

Studies indicate that IL-1β levels are higher in children with MPP, particularly in SMPP (18). Our study found lower serum IL-1β in SMPP and PE patients with poor prognosis than in other groups and identified IL-1β as a protective factor for SMPP. During the inflammatory response, there are complex interactions and negative feedback regulatory mechanisms among multiple cytokines. For example, IL-6 and transforming growth factor β can downregulate and limit the entry of neutrophils into local acute inflammatory sites. This negative feedback mechanism may also play a role in ycoplasma pneumonia, leading to a decrease in IL-1β levels (6, 19). During mycoplasma infection, the intensity and duration of the inflammatory response are regulated by multiple factors. In some cases, IL-1β levels may increase in the early stage of infection; however, as the condition progresses, their changes can be influenced by disease progression, immune system regulation, and the effects of medications [20]. These inconsistent conclusions also indicate the complexity and dynamic nature of the immune mechanisms in mycoplasma pneumonia.

The level of IL-8 was lower in the SMPP group, whereas a study by Deng et al. showed elevated IL-8 levels in the BALF of MPP patients (7). A study by Yang et al. (6, 19) indicated that lung epithelial cells secrete IL-8 following MP infection. This discrepancy may be attributed to differences in detection sites: IL-8 may be upregulated locally in the lungs (in BALF) but suppressed systemically due to negative feedback mechanisms (such as the anti-inflammatory effect of IL-10). Additionally, numerous studies have reported elevated IL-8 levels in the blood. Neutrophil chemotaxis after mycoplasma infection has been associated with SMPP; however, from a broader perspective, there are multiple neutrophil chemotactic factors. Our data suggest that SMPP has little correlation with IL-8, indicating that other key factors responsible for neutrophil chemotaxis and maturation may exist.

IL-10 has a strong anti-inflammatory effect. It suppresses Th1 cell activation and initiates humoral immunity to clear extracellular microbial pathogens. IL-17, a key pro-inflammatory cytokine, recruits neutrophils and boosts IL-1β, IL-6, and TNF-α expression. It is related to the pathological process of MPP (19). Animal studies have shown that IL-17 mediates lung injury by promoting neutrophil aggregation, thereby leading to pleuropneumonia (20). Our results revealed higher serum IL-10 and IL-17 levels in the SMPP and RMPP groups. Prior studies showed significantly higher IL-10 and IL-17 in the SMPP group than in the MMPP group, consistent with our findings (21, 22). This indicates that IL-10 and IL-17 are associated with the severity and prognosis of MPP.

Additionally, our study revealed that IFN-γ, IL-6, and IL-10 were positively correlated with CRP, NLR, D-dimer, and LDH. IL-5, IL-17, and IL-1β were positively correlated with LDH. Meanwhile, CRP, NLR, and LDH are key inflammatory biomarkers positively correlated with the severity of inflammation. As confirmed in multiple studies, CRP, LDH, and NLR can even serve as independent risk factors for RMPP. D-dimer reflects the pathological role of the fibrinolysis and coagulation systems in lung injury, and elevated D-dimer is a risk factor for RMPP (23). ROC analysis revealed that the AUC of IFN-γ, IL-4, IL-5, IL-1β, and IL-6 were 0.53, 0.578, 0.50, 0.536, and 0.607, respectively. Although the results of ROC analysis cannot effectively predict the outcomes of mycoplasma infection, they are helpful for further understanding the pathogenesis.

This study is subject to several limitations. First, as a single-center study, it may be prone to selection bias. In addition, given its retrospective nature, we regret that we were unable to dynamically assess the relationship between cytokines and changes in disease status. Furthermore, we acknowledge that immunomodulators administered before and after cytokine detection—such as corticosteroids, intravenous immunoglobulin, and macrolides—do exert an impact on cytokine levels, and we regret that data pertaining to these factors were not included in our analysis. That said, it is important to contextualize

our findings: all data were collected from specimens obtained within 24 h of admission. While we cannot definitively determine the exact day of the disease course, all measurements were taken during the acute phase of illness. Moreover, outcomes were assessed after admission, so the role of acute-phase inflammatory factors in outcome prediction is deemed valid. To our knowledge, intravenous immunoglobulin therapy was not administered to our patients during their outpatient care prior to hospitalization. We recognize that the use of hormones and antibiotics is indeed critical for exploring the dynamic relationship between cytokine fluctuations and disease progression; this is an important point we plan to address in future prospective studies, which will focus on the dynamic changes in cytokines and mycoplasma pneumonia, as well as the impact of drug administration on cytokine levels.

## Conclusion

Patients with a poor MPP prognosis showed a more intense immunoinflammatory response in the lungs. Abnormal elevations in IL-4, IL-6, IL-10, IL-17, and IFN-γ, along with decreased levels of IL-5 and IL-1β, offer significant insights into disease progression. Integrating cytokine profiles into monitoring and diagnostic protocols can facilitate the timely identification and management of adverse prognoses in MPP. Additionally, our study provides valuable data for elucidating the mechanisms of inflammatory cytokines in MPP.

## MATERIALS AND METHODS

Patients: A cohort study was conducted on 2,391 hospitalized patients with MPP in the Department of Respiratory at the Children's Hospital of Nanjing Medical University from 1 January 2021 to 1 April 2024. Demographic data, clinical characteristics, laboratory test results, chest imaging findings, and prognosis information were retrospectively collected via an electronic medical record platform.

### Inclusion and exclusion criteria

Inclusion criteria: (i) Aged ≥28 d and <18 years; (ii) Symptoms: fever and respiratory symptoms; (iii) Imaging, nucleic acid, and serum antibodies positive, a definite diagnosis of mycoplasma pneumonia. Exclusion criteria: (i) Disease course >4 weeks; (ii) presence of other respiratory diseases such as asthma, tuberculosis, cystic fibrosis, primary ciliary dyskinesia, lung tumors, or interstitial lung diseases; (iii) primary immunodeficiencies, congenital heart disease, or genetic metabolic disorders; (iv) co-infections; (v) incomplete data.

### Data collection

Clinical information was collected from the electronic medical records, including age, sex, clinical manifestations, symptoms, extrapulmonary complications, presence or absence of fever, the duration of fever before admission, length of hospital stay, and chest imaging results. Chest computed tomography (CT) is mainly used when (i) clinical findings do not match chest radiographs, (ii) airway and lung malformations are suspected, (iii) serious complications have occurred, or (iv) the patient does not respond to treatment or has other conditions that need to be ruled out. Laboratory tests at admission include WBC count, NLR, CRP, platelet count, LDH, APTT, prothrombin time (PT), D-dimer, fibrinogen, and levels of 12 inflammatory cytokines (IFN-γ, IFN-α, IL-1β, IL-2, IL-4, IL-5, IL-6, IL-8, IL-10, IL-12p70, IL-17, and TNF-α). The specific detection methods of cytokines are provided in the supplementary materials (Text S1).

### Clinical outcome

The prognoses of MPP obtained from the electronic medical record system were as follows: (i) SMPP: Patients with SMPP met the criteria for severe CAP. (ii) RMPP: Patients

were diagnosed with RMPP if, after 7 days or more of conventional macrolide antibiotic treatment, they still had fever, worsening clinical symptoms, abnormal pulmonary imaging findings, and extrapulmonary sequelae. (iii) PE: Patients were diagnosed with PE through CT pulmonary angiography. (iv) NP: Chest CT showed multiple low-density, thin-walled cavities containing gas or fluid, which resulted from lung consolidation.

## Statistical analysis

Statistical analysis was conducted using SPSS 25.0 (IBM, Armonk, NY, USA). Normally distributed continuous data were described as mean ± SD and compared between groups using Student's $t$-test. Skewed continuous data were presented as M (P25, P75) and compared via the Mann–Whitney U test. Categorical data were expressed as frequency and percentage and compared between groups using the χ test. Logistic regression analyses were conducted to explore influencing factors. ROC curve analysis was performed for diagnostic value. R was used for Spearman's correlation analysis, cluster analysis, and PCA. Statistical significance was set at two-sided $P < 0.05$.

## ACKNOWLEDGMENTS

The authors thank all nursing staff working in our department for keeping extremely detailed patient records, which contributed greatly to the completion of this research.

Feng Liu and Deyu Zhao contributed to the study design. Chang Shu and Anni Cao contributed to data acquisition. Yuling Bao and Rui Zhang analyzed data. Yuling Bao wrote the manuscript, and all authors read and approved the final manuscript.

## AUTHOR AFFILIATION

[1]Department of Respiratory, Children's Hospital of Nanjing Medical University, Nanjing, China

## AUTHOR ORCIDs

Feng Liu  http://orcid.org/0000-0003-2875-1909

## FUNDING

| Funder | Grant(s) | Author(s) |
| --- | --- | --- |
| National Nature Science Foundation of China General Program | 82572609 | Feng Liu |

## DATA AVAILABILITY

The data that support the findings of this study are available from the corresponding author upon reasonable request.

## ETHICS APPROVAL

This study was approved by the Institutional Ethics Committee of the Children's Hospital of Nanjing Medical University (approval number 202310005-1). All procedures performed in this study involving human participants were in accordance with the Declaration of Helsinki (as revised in 2013), and the informed consent of the parents or legal guardians of all enrolled children was obtained.

## ADDITIONAL FILES

The following material is available online.

## Supplemental Material

**Supplemental material (Spectrum01615-25-s0001.docx).** Fig. S1 and S2; Tables S1 and S2.

## Open Peer Review

**PEER REVIEW HISTORY (review-history.pdf).** An accounting of the reviewer comments and feedback.

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
