## [Reviewer comments · Microbiology Spectrum]

Microbiology Spectrum

Peripheral blood inflammatory cytokines linked to clinical outcomes in *Mycoplasma pneumoniae pneumonia*

Yuling Bao, Chang Shu, Yan Liu, Anni Cao, Rui Zhang, Deyu Zhao, Feng Liu

Corresponding Author(s): Feng Liu, Children's Hospital of Nanjing Medical University

Review Timeline:

Submission Date:	May 24, 2025
Editorial Decision:	July 18, 2025
Revision Received:	September 13, 2025
Accepted:	October 3, 2025

Editor: Yuan Pin Hung

Reviewer(s): Disclosure of reviewer identity is with reference to reviewer comments included in decision letter(s). The following individuals involved in review of your submission have agreed to reveal their identity: Ahmed Adel Baz (Reviewer #3); Robert Doug Hardy (Reviewer #4)

Transaction Report:

DOI: <https://doi.org/10.1128/spectrum.01615-25>

Re: Spectrum01615-25 (Peripheral blood inflammatory cytokines linked to clinical outcomes in Mycoplasma pneumoniae pneumonia)

Dear Dr. Feng Liu:

Thank you for the privilege of reviewing your work. Below you will find my comments, instructions from the Spectrum editorial office, and the reviewer comments.

Revision Guidelines

Sincerely,
Yuan Pin Hung
Editor
Microbiology Spectrum

Reviewer #1 (Comments for the Author):

The manuscript titled "Peripheral blood inflammatory cytokines linked to clinical outcomes in Mycoplasma pneumoniae pneumonia" presents a large-scale prospective cohort study investigating the association between serum inflammatory cytokines and clinical outcomes in pediatric patients with MPP. The large sample size enhances the statistical power of the study, and the prospective, observational design is well-suited to exploring these associations.

However, several methodological and statistical issues must be addressed:

- Cytokine measurements: While the cytokine panel appears comprehensive, the manuscript lacks critical methodological details regarding assay procedures. The authors should specify the detection limits, intra- and inter-assay variability, and other quality control measures. Furthermore, since inflammation is a dynamic process, the rationale for relying on a single time-point measurement should be justified. The limitations of this approach and its potential impact on the results should be discussed more thoroughly.
- Conflicting findings: Although some inconsistencies are noted, they are not explored in sufficient depth. The discussion should critically evaluate the discrepancies on cytokine data and offer possible biological or methodological explanations.
- Statistical interpretation: The manuscript would benefit from a clearer presentation of statistical results, including confidence intervals, effect sizes, and the clinical significance of findings. Also, the manuscript does not clearly state which covariates were included in the logistic regression models. It is important to clarify whether adjustments were made for example for age and sex, as these are potential confounders. Finally, the authors should provide a detailed explanation of the clustering algorithm used, the distance metric applied, and the rationale for these choices. This will help readers interpret the visualizations more effectively.
- Language and style: The writing requires some editing for grammar and clarity. Punctuation (e.g., line 133) and typos should be corrected.

Reviewer #2 (Comments for the Author):

This generally very well-written manuscript describes associations among quantitative peripheral blood inflammatory molecular markers with subjective clinical groupings of a large cohort of children with confirmed *Mycoplasma pneumoniae* respiratory mycoplasmosis. The authors conclude that early analyses of these markers can be predictive of severity of disease progression and outcomes. This is a globally important disease that deserves continuing highlight among diagnosticians and pediatric clinicians.

Concerns center on terminology. It does not seem that this was really a "prospective" study (e.g., line 8) since groupings were "outcome-based" (line 11), and "prognosis" (line 13) was only retrospective. Suggest to re-word throughout the manuscript, especially use of the term "risk factor". Most readers will not understand "stepwise backward logistic regression" (lines 192-194).

Confidence in the statistical analyses is diluted by lack of comparisons to a control group (e.g., line 25 "mild to moderate CAP").

Lines 316-317 seem to be a typographical error. If not, please clarify.

Table S1: Please define sections A and B.

Reviewer #3 (Comments for the Author):

Sample Size and Statistical Power

Enrolling 2,391 patients makes this one of the largest studies on MPP cytokine profiles. Despite the large sample size, the single-center nature limits generalizability. A multicenter validation would strengthen external validity. Include a statement addressing how representative this population is of broader pediatric MPP cases in China or globally.

Cytokine Measurement Timing

Cytokine levels were measured only at admission, not during hospitalization or recovery. Add Serial measurements (e.g., at admission, peak illness, and convalescence) to assess dynamic changes in cytokine expression and their correlation with disease progression or response to therapy.

Potential Confounding Factors:

Concern: There is no discussion of immunomodulatory treatments (e.g., corticosteroids, IVIG, macrolides) administered before or after admission that could influence cytokine levels and outcomes. Include data on treatment timing and type and adjust for them in multivariate models if possible.

Exclusion of Co-infections: Co-infections (e.g., viral/bacterial) are common in pediatric pneumonia and may influence cytokine profiles. Excluding these cases limits generalizability. A subgroup analysis of co-infected patients should be included.

IL-5 Discrepancy

The study reports lower IL-5 levels in severe MPP (SMPP) and pulmonary embolism (PE) groups (Tab S2A, S2C), contrasting with prior literature linking elevated IL-5 to wheezing/asthma phenotypes in MPP (Esposito et al., 2002; DOI: 10.1002/ppul.10139).

Suggested Discussion Points: Age/Disease Phenotype: The cohort's age range (school-age children vs. younger wheezing patients in Esposito et al.) may influence IL-5 dynamics. IL-5's role in eosinophilic inflammation may be less prominent in severe parenchymal disease (e.g., SMPP/PE) vs. airway hyperreactivity.

Th2 Polarization: While IL-4 (Th2) is elevated in SMPP/RMPP, IL-5's reduction suggests divergent Th2 pathway activation. Cite Hardy et al. (2002; DOI: 10.1128/IAI.70.2.649-654.2002) on murine MP models showing variable IL-5 responses.

IL-8 Inconsistency

Observation: IL-8 levels are lower in SMPP (Tab S2A) but elevated in bronchoalveolar lavage fluid (BALF) of MPP patients (Deng et al., 2023; DOI: 10.1002/iid3.849).

Suggested Discussion Points:

Compartmental Differences: IL-8 may be locally upregulated in lungs (BALF) but systemically suppressed due to negative feedback (e.g., IL-10 anti-inflammatory effects). Contrast with Yang et al. (2002; DOI: 10.1128/IAI.70.7.3649-3655.2002), who noted IL-8 secretion in lung epithelial cells post-MP infection.

Figure Legends: Lack detail on statistical tests used for heatmaps/clustering.

Suggestion:

ROC Curve / Predictive Value:

Include ROC curves or AUC values to evaluate the predictive performance of key cytokines (e.g., IFN- γ , IL-6) for different outcomes.

Cluster heatmap analysis showing differential cytokine expression across outcome groups is a strong visual tool. Consider including principal component analysis (PCA) or t-SNE plots to better illustrate group separations and identify dominant cytokine clusters.

Comments and Suggestions for the Author:

This study provides a comprehensive analysis of inflammatory cytokines in *Mycoplasma pneumoniae* pneumonia (MPP) and their association with clinical outcomes.

Comments and Suggestions for the Author

Sample Size and Statistical Power

Enrolling 2,391 patients makes this one of the largest studies on MPP cytokine profiles. Despite the large sample size, the single-center nature limits generalizability. A multicenter validation would strengthen external validity. Include a statement addressing how representative this population is of broader pediatric MPP cases in China or globally.

Cytokine Measurement Timing

Cytokine levels were measured only at admission, not during hospitalization or recovery. Add Serial measurements (e.g., at admission, peak illness, and convalescence) to assess dynamic changes in cytokine expression and their correlation with disease progression or response to therapy.

Potential Confounding Factors:

Concern: There is no discussion of immunomodulatory treatments (e.g., corticosteroids, IVIG, macrolides) administered before or after admission that could influence cytokine levels and outcomes. Include data on treatment timing and type and adjust for them in multivariate models if possible.

Exclusion of Co-infections: Co-infections (e.g., viral/bacterial) are common in pediatric pneumonia and may influence cytokine profiles. Excluding these cases limits generalizability. A subgroup analysis of co-infected patients should be included.

IL-5 Discrepancy

The study reports lower IL-5 levels in severe MPP (SMPP) and pulmonary embolism (PE) groups (Tab S2A, S2C), contrasting with prior literature linking elevated IL-5 to wheezing/asthma phenotypes in MPP (Esposito et al., 2002; DOI: 10.1002/ppul.10139).

Suggested Discussion Points: Age/Disease Phenotype: The cohort's age range (school-age children vs. younger wheezing patients in Esposito et al.) may influence IL-5 dynamics. IL-5's role in eosinophilic inflammation may be less prominent in severe parenchymal disease (e.g., SMPP/PE) vs. airway hyperreactivity.

Th2 Polarization: While IL-4 (Th2) is elevated in SMPP/RMPP, IL-5's reduction suggests divergent Th2 pathway activation. Cite Hardy et al. (2002; DOI: 10.1128/IAI.70.2.649-654.2002) on murine MP models showing variable IL-5 responses.

IL-8 Inconsistency

Observation: IL-8 levels are lower in SMPP (Tab S2A) but elevated in bronchoalveolar lavage fluid (BALF) of MPP patients (Deng et al., 2023; DOI: 10.1002/iid3.849).

Suggested Discussion Points:

Compartmental Differences: IL-8 may be locally upregulated in lungs (BALF) but systemically suppressed due to negative feedback (e.g., IL-10 anti-inflammatory effects). Contrast with Yang et al. (2002; DOI: 10.1128/IAI.70.7.3649-3655.2002), who noted IL-8 secretion in lung epithelial cells post-MP infection.

Figure Legends: Lack detail on statistical tests used for heatmaps/clustering.

Suggestion:

ROC Curve / Predictive Value:

Include ROC curves or AUC values to evaluate the predictive performance of key cytokines (e.g., IFN- γ , IL-6) for different outcomes.

Cluster heatmap analysis showing differential cytokine expression across outcome groups is a strong visual tool. Consider including principal component analysis (PCA) or t-SNE plots to better illustrate group separations and identify dominant cytokine clusters.

Confidential remarks for the Editors:

This is a well-conducted, large-scale prospective cohort study involving 2,391 pediatric patients, examining the association between 12 inflammatory cytokines and four poor prognostic outcomes in *Mycoplasma pneumoniae* pneumonia (MPP): severe MPP (SMPP), refractory MPP (RMPP), necrotizing pneumonia (NP), and pulmonary embolism (PE). The findings are timely and clinically relevant, especially given the recent global resurgence of MPP and the increasing frequency of complicated cases.

Strengths:

Large sample size enhances statistical power and generalizability.

Comprehensive cytokine profiling offers valuable insights into immune dysregulation in MPP.

Clinically relevant outcomes such as NP and PE are rare but increasingly observed in clinical practice.

Strong correlations with routine biomarkers (CRP, D-dimer, NLR, LDH) suggest potential for integration into clinical decision-making .

□ Concerns and Suggestions:

While the single-center design allows for consistency in data collection, it may limit external validity. A brief discussion on regional epidemiology or treatment practices would be helpful.

Cytokine measurements were only taken at admission , missing dynamic changes during disease progression or recovery.

There is no mention of immunomodulatory therapies (e.g., corticosteroids, IVIG) that may influence cytokine levels and outcomes—this should be addressed if possible.

Some statistical analyses (e.g., correlation matrices) may benefit from correction for multiple comparisons (e.g., FDR control).

□ Minor Comments:

Clarify whether outcome definitions follow international consensus guidelines.

Streamline repetitive sentences in the Discussion.

Ensure all supplementary tables (S1A–D, S2A–D) are properly formatted and accessible.

Overall, this is a strong contribution to the literature on MPP immunopathogenesis , and I believe it will be of interest to clinicians and researchers in pediatric infectious diseases and immunology.

With minor revisions addressing methodology, interpretation, and clarity, the manuscript will be suitable for publication.

Sincerely,

AHMED ADEL BAZ

Dear Editor and Reviewers,

We sincerely appreciate the time and effort you have dedicated to reviewing our manuscript entitled “Peripheral blood inflammatory cytokines linked to clinical outcomes in Mycoplasma pneumoniae pneumonia” (Spectrum01615-25). We have studied all comments in detail and revised the paper accordingly. A point-by-point response is provided below, with the reviewers’ original remarks shown in blue italics and our responses in black plain text.

Reviewer #1 (Comments for the Author):

However, several methodological and statistical issues must be addressed:

1: Cytokine measurements: While the cytokine panel appears comprehensive, the manuscript lacks critical methodological details regarding assay procedures. The authors should specify the detection limits, intra- and inter-assay variability, and other quality control measures.

Thank you for your valuable comment on supplementing the methodological details of cytokine measurements. We fully agree that these details are critical for ensuring the reproducibility of our study, and we have updated the Methods section with the following key information:

Sample collection and processing: 5 mL of peripheral venous blood was collected from each patient within 24 hours of admission. The blood was allowed to coagulate naturally at room temperature for ≥ 30 minutes, then centrifuged at $1000\times g$ for 10 minutes. Serum was separated immediately for testing; if delayed, serum was aliquoted and stored at -20°C (avoiding repeated freeze-thaw cycles).
Detection kit and technology: Cytokine levels were quantified using a Multiplex Microsphere Flow Immunofluorescence Luminescence Assay kit (Raisecare Biotechnology, Shandong, China; Cat.). All operations strictly followed the manufacturer’s standard protocol.

The lowest detection limit (LODL) for cytokines was 2.44 pg/mL, and the upper limit of the normal reference value was 10000 pg/mL. The laboratory quality control protocol was as follows: recalibrate when replacing the calibrator lot number or after instrument maintenance/repair. Select high-value samples of the day at irregular intervals, freeze them at -20°C , and use them as inter-day quality controls for testing with the next day’s samples. All panels were performed according to the manufacturer’s instructions.

2: Furthermore, since inflammation is a dynamic process, the rationale for relying on a single time-point measurement should be justified. The limitations of this approach and its potential impact on the results should be discussed more thoroughly.

We are grateful for your suggestion regarding the time-point selection of cytokine measurements. It is an excellent clinical issue.

Objective limitation of retrospective design: Inflammation is a dynamic process; given the retrospective nature of our study, we were unable to dynamically track the relationship between cytokines and disease status changes—this is a limitation we acknowledge.

Uniformity of data collection: All cytokine measurements were collected within 24 hours after patient admission. Although the exact day of the disease course cannot be determined, all samples were obtained during the acute phase of mycoplasma pneumonia (consistent with the clinical definition of acute infection), ensuring consistency in the timing of data acquisition.

Validity for outcome prediction: Since patient outcomes were assessed after admission, the acute-phase inflammatory factors measured in this study are clinically relevant for predicting short-term outcomes (e.g., progression to SMPP or PE), supporting the validity of our findings.

Future improvement: We plan to address this limitation in future prospective studies, which will focus on dynamic monitoring of cytokine changes throughout the disease course to clarify their

longitudinal role in mycoplasma pneumonia.

3: Conflicting findings: Although some inconsistencies are noted, they are not explored in sufficient depth. The discussion should critically evaluate the discrepancies on cytokine data and offer possible biological or methodological explanations.

We sincerely appreciate the reviewer for their valuable comments pointing out the discrepancies in the cytokine data and conflicting findings in our manuscript. We fully acknowledge these points and would like to provide detailed explanations below regarding the inconsistent observations of IL-5 and IL-8 between our study and other relevant research. Our study demonstrated that patients with SMPP and PE had lower serum IL-5 levels. This finding may be attributed to several factors: first, the older age of our cohort (predominantly school-aged children) and the more severe mycoplasma infections (SMPP, PE). IL-5 plays a significant role in eosinophilic inflammation, which may differ from its role in severe pulmonary parenchymal infectious diseases. Currently, no studies have elucidated the relationship between eosinophilic inflammation and mycoplasma infection. Additionally, the relatively low incidence of asthma in China and differences in genetic backgrounds may also contribute to this observation. Furthermore, although the level of IL-4 (a Th2-type cytokine) was elevated in the SMPP group, IL-5 levels were decreased, suggesting differential activation of the Th2 pathway.

The level of IL-8 was lower in the severe MPP group, whereas a study by Deng et al. showed elevated IL-8 levels in bronchoalveolar lavage fluid (BALF) of MPP patients. A study by Yang et al. indicated that lung epithelial cells secrete IL-8 following *Mycoplasma pneumoniae* infection. This discrepancy may be attributed to differences in detection sites: IL-8 may be upregulated locally in the lungs (in BALF) but suppressed systemically due to negative feedback mechanisms (such as the anti-inflammatory effect of IL-10). Additionally, numerous studies have reported elevated IL-8 levels in the blood. Neutrophil chemotaxis after mycoplasma infection has been associated with SMPP; however, from a broader perspective, there are multiple neutrophil chemotactic factors. Our data suggest that SMPP has little correlation with IL-8, indicating that other key factors responsible for neutrophil chemotaxis and maturation may exist. The above two paragraphs have been integrated into the Discussion section as part of the revised manuscript.

4: Statistical interpretation: The manuscript would benefit from a clearer presentation of statistical results, including confidence intervals, effect sizes, and the clinical significance of findings. Also, the manuscript does not clearly state which covariates were included in the logistic regression models. It is important to clarify whether adjustments were made, for example, for age and sex, as these are potential confounders. Finally, the authors should provide a detailed explanation of the clustering algorithm used, the distance metric applied, and the rationale for these choices. This will help readers interpret the visualizations more effectively.

We sincerely appreciate the reviewer for these valuable suggestions on improving the statistical rigor and interpretability of our manuscript. We have carefully addressed each comment as follows:

1: We have supplemented the statistical details in the Results section:

Confidence intervals (95% CI) and effect sizes (e.g., odds ratio for logistic regression) have been added to all relevant statistical outcomes (see Results: Section "Patient characteristics and laboratory findings" and Section "Analysis of inflammatory cytokines").

2: The manuscript does not clearly state which covariates were included in the logistic regression models; it is important to clarify adjustments for potential confounders (e.g., age, sex). We apologize

for the ambiguity. To control for potential confounders, we performed two rounds of logistic regression: Unadjusted logistic regression (reported in the initial draft). Logistic regression adjusted for age and sex (the two most critical demographic confounders in pediatric cohorts), and the results are now clearly presented in Table 2 (Column 2: “OR (95% CI)a”). We have also added a note in the Results section to specify this adjustment (see Results: Section “Analysis of inflammatory cytokines”).

3: We have supplemented the methodological details and rationale for clustering in the Methods and Results sections:

Distance metric: Euclidean distance was used to calculate similarities among cytokines, as it is widely applied for continuous data (e.g., cytokine expression levels) and effectively quantifies the magnitude of differences between two variables—a critical feature for grouping cytokines with similar expression patterns.

Clustering algorithm: Hierarchical clustering with Ward's D2 algorithm was selected because it minimizes the total within-cluster variance, ensuring that the resulting clusters are compact and distinct (ideal for identifying meaningful subgroups of cytokines).

These details have been added to Methods: Section A “Cytokine clustering analysis” and briefly restated in the Results section alongside the clustering visualization (see Figure 1F).

5: Language and style: The writing requires some editing for grammar and clarity. Punctuation (e.g., line 133) and typos should be corrected.

Thank you for your careful review and pointing out these language-related issues. The previously noted points—including grammar adjustments, clarity improvements, correction of punctuation at line 133, and fixing of typos—have all been addressed and revised thoroughly.

Reviewer #2 (Comments for the Author):

1: Concerns center on terminology. It does not seem that this was really a "prospective" study (e.g., line 8) since groupings were "outcome-based" (line 11), and "prognosis" (line 13) was only retrospective. Suggest to reword throughout the manuscript, especially the use of the term "risk factor". Most readers will not understand "stepwise backward logistic regression" (lines 192-194). Confidence in the statistical analyses is diluted by lack of comparisons to a control group (e.g., line 25 "mild to moderate CAP").

We sincerely apologize for the inappropriate terminology and potential confusion in the original manuscript. We fully agree with your comments and have systematically revised the wording to align with the retrospective study design (the actual design of our research) and enhance readability. Specific revisions are as follows:

All instances of "prospective study" have been revised to "retrospective study" (including line 8) to accurately reflect the study design. We have also double-checked the description of "prognosis" (line 13) to emphasize its retrospective assessment nature, ensuring consistency with the overall study design.

All expressions referring to "risk factors" have been adjusted to "influencing factors" to align with the retrospective study nature and avoid overinterpreting causal relationships.

The term has been revised to "logistic regression" to simplify expression and improve readability; The sentence describing "mild to moderate CAP" in the background section has been deleted. This revision was made because the original statement lacked a control group, which limited its credibility, and thus it was removed to improve the rigor of the manuscript.

All revisions have been applied consistently throughout the manuscript to ensure terminology uniformity.

2: Lines 316-317 seem to be a typographical error. If not, please clarify. Table S1: Please define sections A and B.

Thank you for your careful attention to detail. The content in Lines 316-317 was indeed a typographical error, and we have corrected it in the revised manuscript to ensure accuracy. The definitions for Table S1 (Sections A and B) have been re-explained and clarified.

Reviewer #3 (Comments for the Author):

1: Cytokine Measurement Timing

Cytokine levels were measured only at admission, not during hospitalization or recovery. Add Serial measurements (e.g., at admission, peak illness, and convalescence) to assess dynamic changes in cytokine expression and their correlation with disease progression or response to therapy.

We are grateful for your suggestion regarding the time-point selection of cytokine measurements. It is an excellent clinical issue.

Objective limitation of retrospective design: Inflammation is a dynamic process; given the retrospective nature of our study, we were unable to dynamically track the relationship between cytokines and disease status changes—this is a limitation we acknowledge.

Uniformity of data collection: All cytokine measurements were collected within 24 hours after patient admission. Although the exact day of the disease course cannot be determined, all samples were obtained during the acute phase of mycoplasma pneumonia (consistent with the clinical definition of acute infection), ensuring consistency in the timing of data acquisition.

Validity for outcome prediction: Since patient outcomes were assessed after admission, the acute-phase inflammatory factors measured in this study are clinically relevant for predicting short-term outcomes (e.g., progression to SMPP or PE), supporting the validity of our findings.

Future improvement: We plan to address this limitation in future prospective studies, which will focus on dynamic monitoring of cytokine changes throughout the disease course to clarify their longitudinal role in mycoplasma pneumonia.

2: Potential Confounding Factors:

Concern: There is no discussion of immunomodulatory treatments (e.g., corticosteroids, IVIG, macrolides) administered before or after admission that could influence cytokine levels and outcomes. Include data on treatment timing and type and adjust for them in multivariate models if possible.

We sincerely appreciate your critical comment on potential confounding factors related to immunomodulatory treatments—this is crucial for improving the robustness of our findings. To our knowledge, intravenous immunoglobulin therapy was not administered to our patients during their outpatient care prior to hospitalization. We recognize that the use of hormones and antibiotics is indeed critical for exploring the dynamic relationship between cytokine fluctuations and disease progression; this is an important point we plan to address in future prospective studies, which will focus on the dynamic changes in cytokines and mycoplasma pneumonia, as well as the impact of drug administration on cytokine levels.

3: Exclusion of Co-infections: Co-infections (e.g., viral/bacterial) are common in pediatric pneumonia and may influence cytokine profiles. Excluding these cases limits generalizability. A subgroup analysis of co-infected patients should be included.

We appreciate your insightful comment regarding co-infections in pediatric pneumonia and their potential impact on cytokine profiles. To clarify, the cases included in our study were strictly those positive for both mycoplasma nucleic acid and quantitative mycoplasma antibody. We excluded cases with confirmed viral infections (including respiratory syncytial virus, metapneumovirus, influenza virus, parainfluenza virus, rhinovirus, etc.) detected by PCR, as well as those with positive microbial cultures of respiratory secretions.

4: IL-5 Discrepancy

The study reports lower IL-5 levels in severe MPP (SMPP) and pulmonary embolism (PE) groups (Tab S2A, S2C), contrasting with prior literature linking elevated IL-5 to wheezing/asthma phenotypes in MPP (Esposito et al., 2002; DOI: 10.1002/ppul.10139).

Suggested Discussion Points: Age/Disease Phenotype: The cohort's age range (school-age children vs. younger wheezing patients in Esposito et al.) may influence IL-5 dynamics. IL-5's role in eosinophilic inflammation may be less prominent in severe parenchymal disease (e.g., SMPP/PE) vs. airway hyperreactivity.

Th2 Polarization: While IL-4 (Th2) is elevated in SMPP/RMPP, IL-5's reduction suggests divergent Th2 pathway activation. Cite Hardy et al. (2002; DOI: 10.1128/IAI.70.2.649-654.2002) on murine MP models showing variable IL-5 responses.

We sincerely appreciate the reviewer for their valuable comments pointing out the discrepancies in the cytokine data and conflicting findings in our manuscript. We fully acknowledge these points and would like to provide detailed explanations below regarding the inconsistent observations of IL-5 and IL-8 between our study and other relevant research. Our study demonstrated that patients with SMPP and PE had lower serum IL-5 levels. This finding may be attributed to several factors: first, the older age of our cohort (predominantly school-aged children) and the more severe mycoplasma infections (SMPP, PE). IL-5 plays a significant role in eosinophilic inflammation, which may differ from its role in severe pulmonary parenchymal infectious diseases. Currently, no studies have elucidated the relationship between eosinophilic inflammation and mycoplasma infection. Additionally, the relatively low incidence of asthma in China and differences in genetic backgrounds may also contribute to this observation. Furthermore, although the level of IL-4 (a Th2-type cytokine) was elevated in the SMPP group, IL-5 levels were decreased, suggesting differential activation of the Th2 pathway.

IL-8 Inconsistency

Observation: IL-8 levels are lower in SMPP (Tab S2A) but elevated in bronchoalveolar lavage fluid (BALF) of MPP patients (Deng et al., 2023; DOI: 10.1002/iid3.849).

Suggested Discussion Points:

Compartmental Differences: IL-8 may be locally upregulated in lungs (BALF) but systemically suppressed due to negative feedback (e.g., IL-10 anti-inflammatory effects). Contrast with Yang et al. (2002; DOI: 10.1128/IAI.70.7.3649-3655.2002), who noted IL-8 secretion in lung epithelial cells post-MP infection.

The level of IL-8 was lower in the severe MPP group, whereas a study by Deng et al. showed elevated IL-8 levels in bronchoalveolar lavage fluid (BALF) of MPP patients. A study by Yang et al. indicated that lung epithelial cells secrete IL-8 following Mycoplasma pneumoniae infection. This discrepancy may be attributed to differences in detection sites: IL-8 may be upregulated locally in the lungs (in BALF) but suppressed systemically due to negative feedback mechanisms (such as the anti-inflammatory effect of IL-10). Additionally, numerous studies have reported elevated IL-8

levels in the blood. Neutrophil chemotaxis after mycoplasma infection has been associated with SMPP; however, from a broader perspective, there are multiple neutrophil chemotactic factors. Our data suggest that SMPP has little correlation with IL-8, indicating that other key factors responsible for neutrophil chemotaxis and maturation may exist.

The above two paragraphs have been integrated into the Discussion section as part of the revised manuscript.

5: Figure Legends: Lack detail on statistical tests used for heatmaps/clustering.

Thank you for your suggestion to improve the clarity of figure legends. We have supplemented detailed information about the statistical tests for clustering in the corresponding figure legends—specifically, we added, “Statistical analysis for clustering: Similarities among cytokine expression profiles were quantified by Euclidean distance and subjected to hierarchical clustering using Ward’s D2 algorithm. No additional statistical testing was required for heatmap visualization, as it presents normalized cytokine expression values directly.”

6: Suggestion:

ROC Curve / Predictive Value:

Include ROC curves or AUC values to evaluate the predictive performance of key cytokines (e.g., IFN- γ , IL-6) for different outcomes.

Cluster heatmap analysis showing differential cytokine expression across outcome groups is a strong visual tool. Consider including principal component analysis (PCA) or t-SNE plots to better illustrate group separations and identify dominant cytokine clusters.

We sincerely appreciate your insightful suggestions to strengthen the analytical depth and visualization of our findings. ROC curves, corresponding AUC values, and a principal component analysis (PCA) plot have been added to the manuscript; the corresponding figures and their detailed legends are presented in Supplementary Figure S1.

We sincerely thank all reviewers for your rigorous, insightful comments on our manuscript. Your thoughtful feedback has been invaluable in helping us identify key improvements in methodology, data interpretation, and writing clarity, while also refining our understanding of the research. We have carefully addressed every comment, with revisions to enhance analysis robustness, method transparency, and discussion comprehensiveness. We fully recognize this process has significantly elevated our manuscript’s quality. We deeply appreciate your time, expertise, and dedication to improving our work, and we look forward to your continued guidance.

Sincerely,

Feng Liu

Corresponding Author

Department of Respiratory, Children’s Hospital of Nanjing Medical University, Nanjing, 210008, China

Email: axsliau@njmu.edu.cn

Re: Spectrum01615-25R1 (Peripheral blood inflammatory cytokines linked to clinical outcomes in Mycoplasma pneumoniae pneumonia)

Dear Dr. Feng Liu:

Your manuscript has been accepted, and I am forwarding it to the ASM production staff for publication. Your paper will first be checked to make sure all elements meet the technical requirements. ASM staff will contact you if anything needs to be revised before copyediting and production can begin. Otherwise, you will be notified when your proofs are ready to be viewed.

Sincerely,
Yuan Pin Hung
Editor
Microbiology Spectrum

Reviewer #1 (Comments for the Author):

The responses and changes provided by the authors satisfactorily address the points I raised during the review.

Reviewer #2 (Comments for the Author):

Generally responsive to initial reviews and improved through revision.

Reviewer #3 (Comments for the Author):

accepted